# TextSETTR: Label-Free Text Style Extraction and Tunable Targeted Restyling

## Abstract

We present a novel approach to the problem of text style transfer. Unlike previous approaches that use parallel or non-parallel labeled data, our technique removes the need for labels entirely, relying instead on the implicit connection in style between adjacent sentences in unlabeled text. We show that T5 (Raffel et al., 2020), a strong pretrained text-to-text model, can be adapted to extract a style vector from arbitrary text and use this vector to condition the decoder to perform style transfer. As the resulting learned style vector space encodes many facets of textual style, we recast transfers as "targeted restyling" vector operations that adjust specific attributes of the input text while preserving others. When trained over unlabeled Amazon reviews data, our resulting TextSETTR model is competitive on sentiment transfer, even when given only four exemplars of each class. Furthermore, we demonstrate that a single model trained on unlabeled Common Crawl data is capable of transferring along multiple dimensions including dialect, emotiveness, formality, politeness, and sentiment.

## 1 Introduction

There has been a recent surge of interest in text style transfer, with the aim of training models able to modify specific attributes of input text (e.g., sentiment or formality) while preserving the remaining content. For example, a sentiment transfer model might transform the input "best book ever!" into "worst book ever!", while a formality transfer model might change the same input into "This is the best book I have ever read."

Work in this area falls into three categories. **Supervised** approaches like Jhamtani et al. (2017) transfer between pre-selected styles, and rely on aligned parallel training data to teach the model the desired input/output correspondence. This method is limited by the availability of parallel corpora. So-called **"unsupervised"** approaches like Li et al. (2018) and Lample et al. (2019) remove the requirement for parallel data, but still require labeled training examples of each style, and are limited to transfer between a pre-specified set of styles. **Label-free** approaches like the recent Xu et al. (2020) remove the need for any training labels. While the most technically challenging, this offers the potential for transferring between arbitrary styles at inference time and has significant value, as curated datasets are not available for many style attributes.

In this work, we explore the hypothesis that large pretrained text-to-text models like T5 (Raffel et al., 2020) already contain a strong representation of textual style, which can be extracted and used to condition the decoder of a style transfer model through a relatively lightweight fine-tuning procedure. To isolate style information in the absence of labels, we rely on the observation that style is a "slow-moving" feature, which tends to be consistent over large spans of text. Specifically, given two adjacent sentences from an unlabeled corpus, we train our model to extract a "style vector" from the first and use that vector to perform denoising and other reconstruction tasks on the second. This technique extends the unsupervised approach of Lample et al. (2019) to the label-free setting, and allows us to reformulate the style transfer operation as a directional operation in style vector space using the difference between target and source style vectors; we call this "targeted restyling". When combined with a novel "tunable inference" technique for controlling token add/delete rates, this gives our final model: **Text S**tyle **E**xtraction and **T**unable **T**argeted **R**estyling (TextSETTR).

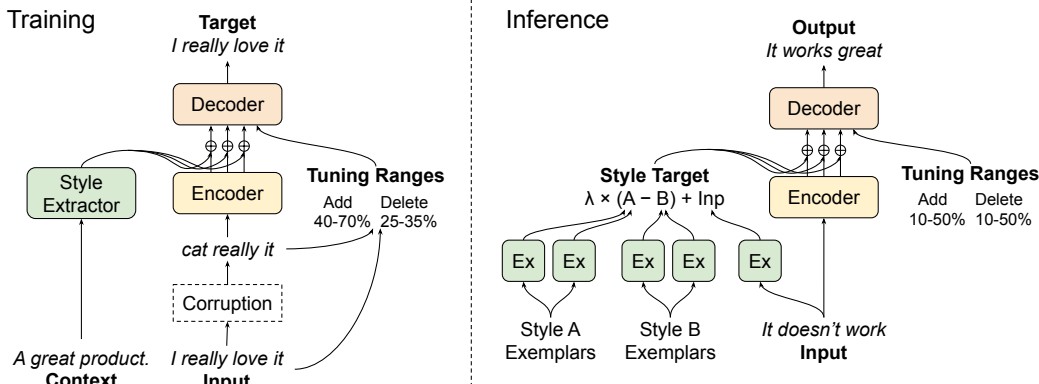

Figure 1: TextSETTR architecture for label-free style transfer. The Encoder, Decoder and Style Extractor (Ex) are transformer stacks initialized from pretrained T5. During training, the model reconstructs a corrupted input, conditioned on a fixed-width "style vector" extracted from the preceding sentence. At inference time, a new style vector is formed via "targeted restyling": adding a directional delta to the extracted style of the input text. Stochastic tuning ranges provide extra conditioning for the decoder, and enable fine-grained control of inference.

Our main contributions are to: (1) demonstrate the viability of label-free style transfer,[1] (2) use sentence adjacency as a means for inducing text style representations, (3) reframe style transfer as "targeted restyling" directional operations in style space, (4) introduce "tunable inference" for finer-grained control of transfers, (5) show the effectiveness of "noisy" back-translation training, and (6) illustrate few-shot generalization to a range of style attributes including dialect, emotiveness, formality, politeness, and sentiment.

## 2 METHOD

Figure 1 illustrates our proposed TextSETTR architecture. At a high level, our approach follows Lample et al. (2019), who train a denoising auto-encoder conditioned on a fixed-width style vector. The key difference in our case is that the true style is unknown at training time. To overcome this, we jointly train a "style extractor" component to induce a useful style representation (that can aid in reconstruction) from text in the nearby context. We describe this in more detail below.

### 2.1 MODEL ARCHITECTURE

We conduct our experiments using a modified version of the Text-to-Text Transfer Transformer (T5) (Raffel et al., 2020). Like T5, our model includes a transformer-based encoder and decoder. As in T5 pretraining, the input to the encoder is a corrupted/noised version of the target, resulting in a reconstruction task. Our goal is to design a type of corruption that results in this training task resembling style transfer, despite the lack of labeled training data.

Our core addition to T5 is the style extractor. Based on the encoder's architecture, this component's input is an uncorrupted sentence in the same style as the target; relying on our assumption that style is a slow-moving feature, we use the sentence preceding the target (the "context") for this.[2] This encourages extracting a style representation that is useful for repairing the corrupted input.

The only architectural difference between the encoder and style extractor is that we mean-pool the style extractor's hidden state sequence into a single fixed-width vector (the "style vector"); in our experiments, the dimensionality of this vector and the encoder hidden states is 1024. To incorporate the style vector into the rest of the model, we simply add it to each of the final encoder hidden states.

---

[1]Our work is concurrent with Xu et al. (2020), who offer a substantially different approach to label-free style transfer, as discussed in Sections 3 and 5.

[2]This approach is similar to the use of adjacent sentences for weak supervision in Devlin et al. (2019) and Zhang et al. (2020).

We initialize the weights of our model with those of a pretrained T5 model. We initialize both the style extractor and encoder from the pretrained encoder, but the weights are *not* tied during training.

## 2.2 CORRUPTION STRATEGIES

We experiment with combinations of three different reconstruction tasks, each contributing a loss term. All three share the same overall structure, where a sentence $s_i$ in the dataset is corrupted by some function $f$ to produce $\tilde{s}_i = f(s_i)$. The cross-entropy loss is calculated using the uncorrupted sentence $s_i$ as the target, the corrupted sentence $\tilde{s}_i$ as the input, and the uncorrupted preceding sentence $s_{i-1}$ as the context. The three choices of $f$ are Noise (N), Back-Translation (BT), and Noisy Back-Translation (NBT), described below.

**Noise (N)** This function corrupts the input by (i) dropping, (ii) replacing, and/or (iii) shuffling tokens, in that order. For each example we sample a separate noise probability $p$ for each sub-type of noise from a uniform distribution in the range 20–60%. We do this because the model should be able to "undo" varying degrees of noise, as some types of style transfer may require changing only a few tokens, while others may require larger rewrites.

For *drop* noise, we drop each token in $s_i$ with independent probability $p$. For *replace* noise, let $s_{ik}$ be the $k$-th token within $s_i$. For each $s_i$, a random other example $s_j$ is chosen, and then each token $s_{ik}$ is replaced by $s_{jk}$ with probability $p$. If $s_j$ has fewer than $k$ tokens, then the replacement does not occur. For *shuffle* noise, each token in $s_i$ is chosen with probability $p$, and then all chosen tokens are randomly shuffled to the position of another chosen token, leaving non-chosen tokens in place.

The use of drop and shuffle noise results in a loss term similar to the denoising loss used by Lample et al. (2019). Their motivation for this loss was to encourage language modeling. As we fine-tune an already-strong T5 language model in our experiments, our motivation is rather to introduce a *conditional* element to the language model, in the form of the extracted style vector input.

**Back-Translation (BT)** This corruption function, used by Lample et al. (2019), runs the current version of the model in inference mode to transfer $s_i$ into a different style, giving the corrupted $\tilde{s}_i$. In prior work using labels, specifying a different target style was straightforward. In our case, because we do not have access to labels, we simply sample a random sentence $s_j$ to use as the context. To increase diversity of the generated examples, we decode with sampling instead of greedy decoding.

Because $\tilde{s}_i$ is produced by a strong language model, BT should result in training examples where both the input and output are coherent sentences, matching our intended inference setting. By contrast, the text produced by "Noise" corruption does not resemble test-time inputs.

**Noisy Back-Translation (NBT)** This novel corruption function is a composition of the previous two. Noise is first applied to $s_i$ as described above, and the result is used as the input (with randomly-sampled $s_j$ as the context) to the model in inference mode to produce $\tilde{s}_i$ via sampling, as in BT.

Once the model has learned to undo random noise from the associated loss term, NBT should produce training examples where some of the tokens are preserved from $s_i$ while others were generated by the model itself under the influence of the "incorrect" context $s_j$. This is similar to BT, but we hypothesize that it may be better suited to style transfer. BT was originally used for machine translation (Sennrich et al., 2016), a setting where most or all input tokens need to be changed. In contrast, style transfer within a single language usually requires only changing a subset of input tokens; the training examples resulting from the NBT procedure should have this property. We believe that this will encourage the model to identify which tokens in the input do not match the target style indicated by $s_{i-1}$ and change them, which is exactly what we want a style transfer model to do. This is conceptually similar to the work of Clark et al. (2020), who pretrain language models to discriminate which input tokens were originally present and which were altered by a simpler language model.

**Final Loss** The final loss term used for training is the sum of the above loss terms, each calculated from the same input $s_i$. However, not every model we experiment with includes all three losses.

## 2.3 INFERENCE PROCEDURE

**Tunable Add/Delete Rates** In preliminary experiments, we observed a recurring problem that the model would often change either far too little (failing to achieve the target style), or far too much

(failing to preserve the input content). To address this problem, we introduce a "tunable inference" mechanism to constrain how much content should be added and deleted at inference time.

For every input/output pair during training, we calculate the proportions of tokens that were added and deleted. The "add rate" is the proportion of output tokens absent from the input, and the "delete rate" is the proportion of input tokens absent from the output.[3] Rather than provide these rates directly to the decoder, we provide *ranges*, covering but not necessarily centered around the true rates. Specifically, we sample each range width uniformly from [0,1], and uniformly sample the "alignment" of the true rate within the range. The final ranges are clipped to [0,1], and passed to the decoder as four values: [min_add_rate, max_add_rate, min_del_rate, max_del_rate].[4] This approach provides more flexibility at inference time, so we can enforce tight or loose constraints on each rate.

**Targeted Restyling** While previous work on style transfer has largely assumed a fixed set of discrete styles, our method can extract a "ready-made" style vector from any sentence. We expect this learned representation to capture a rich summary of the sentence, covering many attributes. For example, a given style vector might encode that a sentence is informal, humorous, in British English, and so on.

In this framework, transferring a single attribute (e.g., informal → formal) is not as simple as just providing a vanilla "formal" style target, as this would ignore all the other attributes that defined the original input. Rather, we must operate in style space to construct a new target style that is simultaneously formal, humorous, British, and so on.

Concretely, at inference time, we assume access to a small set of "exemplar" sentences (between 1 and 100) for both the source value (e.g., informal) and target value (e.g., formal) of the attribute being modified. We infer style vectors for each exemplar using the style extractor, and take the mean of each class, giving vectors $v^{src}$ and $v^{trg}$. Assuming the exemplar pools are relatively diverse, this averaging should "wash out" most attributes not being targeted.

To transfer an input sentence $x$, we apply a targeted restyling in the appropriate direction. After extracting the original style from the input itself, $v^x$, we compute the target output style by moving in the direction of the delta between the source and target attributes values, as in (1), producing the style vector used for decoding. In practice, we find that the delta scale $\lambda$ is an important hyperparameter to tune. Generally values in the range $[1.0, 10.0]$ work well, with the best values depending on the attribute and the exemplars in question.

$$v^x + \lambda \times \left( v^{trg} - v^{src} \right) \tag{1}$$

## 3 EXPERIMENTS

To evaluate our approach and better understand the effects of our various design choices, we test on label-free sentiment transfer, using the Amazon reviews dataset of Li et al. (2018). However, as their training split doesn't indicate which sentences were adjacent in the original reviews, we make use of a different source of raw review text.

**Training Procedure** Our unlabeled training data comes from the 233.1M Amazon reviews provided by Ni et al. (2019). Ignoring the star ratings completely, we extract adjacent lines from multi-line reviews to use as the context and input for our training procedure, giving 23.6M examples. We also preprocess all text to match the format of the Li et al. (2018) data, as detailed in Appendix A.4. Initializing our model from pretrained T5 (t5.1.1.large)[5], we fine-tune on these examples, optimizing the joint reconstruction loss from Section 2. Our default TextSETTR configuration is selected based on preliminary experiments (on development data) varying the set of reconstruction tasks and inference procedures. The model uses an equally weighted combination of the Noise (N) and Noisy Back-Translation (NBT) tasks. For both tasks, we use drop and replace noise, but no shuffle noise. We fine-tune for 10k steps, with a batch size of 65,536 tokens, and a fixed learning rate of 1e-3.

---

[3]This calculation ignores word order. As one example, if a token appears three times in the input and five times in the output, two of the five occurrences are counted as "added".

[4]More specifically, a vector is prepended to the sequence of encoder hidden states, holding the tuning rates as the first four values, and zeros elsewhere.

[5]https://github.com/google-research/text-to-text-transfer-transformer

| Model | Acc. | Content |
|---|---|---|
| TextSETTR | 73.7 | 34.7 |
| N | 23.4 | 84.4 |
| NBT | 70.0 | 27.8 |
| N + BT | 13.3 | 98.7 |
| −replace noise | 66.1 | 42.1 |
| +shuffle noise | 70.3 | 34.1 |
| manual exemplars | 52.4 | 44.2 |
| 1000 exemplars | 74.5 | 37.2 |
| −tunable inference | 71.5 | 39.4 |
| overwrite style | 25.3 | 55.8 |
| small train set | 74.5 | 33.4 |
| CP-G | 51.1 | 35.5 |
| CP-B | 36.3 | 39.8 |
| CrossAligned | 68.2 | 2.9 |
| Delete&Retrieve | 49.4 | 56.9 |
| B-GST | 60.2 | 54.2 |

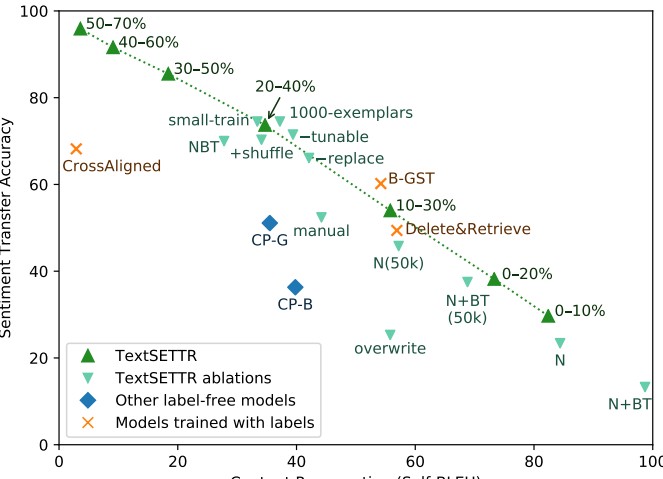

Figure 2: Automatic evaluation metrics comparing our TextSETTR model, ablations, and previous work. Up-and-right is better. We train for 10k steps and use add/delete:20–40% unless otherwise specified. We recalculate metrics for previous approaches, using our BERT classifier for accuracy, ensuring direct comparability with our models.

| Model | Accuracy | Content |
|---|---|---|
| TextSETTR (0–20%) | 63.4 | 76.9 |
| TextSETTR (10–30%) | 72.7 | 60.2 |
| TextSETTR (20–40%) | 83.6 | 39.4 |
| TextSETTR (30–50%) | 89.7 | 21.5 |
| TextSETTR (40–60%) | 94.3 | 11.3 |
| TextSETTR (50–70%) | 96.6 | 5.0 |
| Lample et al. 2019 | 82.6 | 54.8 |

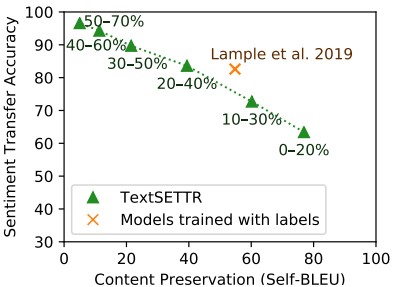

Figure 3: Comparison with Lample et al. (2019) on the evaluation setting that includes pos→pos and neg→neg transfers. Note, a model that simply copies its input can achieve 50% accuracy.

**Evaluation Procedure** Following previous work, we use automatic metrics to assess attribute control (sentiment) and content preservation on the Li et al. (2018) test data. To estimate the sentiment of the transferred output text, we fine-tune a BERT-Large classifier (Devlin et al., 2019) on the Li et al. (2018) train split, scoring 87.8% accuracy on the dev split. For content preservation, we follow Sudhakar et al. (2019) and Xu et al. (2020) and report self-BLEU between the output and input text, calculated using SacreBLEU (Post, 2018)[6]. Some prior work reports instead BLEU scores between outputs and human-generated transfers from Li et al. (2018); we observe this metric to be highly correlated with self-BLEU but report it in Appendix A.3 for completeness.

To perform transfers, we follow the procedure from Section 2.3. For our default setup, we sample 100 positive and 100 negative exemplars from Li et al. (2018) train. We also experiment with (i) increasing to 1000 exemplars of each class, and (ii) decreasing to just four manually selected exemplars of each class. Unless otherwise specified, we use greedy decoding, a delta scale of $\lambda$=8, and add/delete tuning ranges of 20–40%.

**Core Results** Figure 2 shows our core results. Our default TextSETTR configuration (using N+NBT training and tuning ranges 20–40%) achieves 73.7% accuracy at swapping sentiment (as judged by the classifier), while still staying somewhat close to the original input text (self-BLEU 34.7). Due to our tunable inference technique, we can also trade off accuracy for content preservation by adjusting the add/delete rates, as seen in the different points along the green line. Notably, TextSETTR outperforms the label-free CP-G and CP-B models of Xu et al. (2020). More remarkably, TextSETTR

---

[6]Version string: BLEU+case.mixed+numrefs.1+smooth.exp+tok.13a+version.1.4.13

| | Negative $\rightarrow$ Positive | | | Positive $\rightarrow$ Negative | | |
|---|---|---|---|---|---|---|
| Model | Sentiment | Preservation | Fluency | Sentiment | Preservation | Fluency |
| TextSETTR | 2.8 | 2.4 | 4.2 | 2.3 | 2.8 | 3.8 |
| Delete&Retrieve | 2.7 | 2.9 | 3.2 | 2.3 | 3.4 | 3.4 |
| B-GST | 2.3 | 2.8 | 3.6 | 2.1 | 3.0 | 3.6 |

Table 1: Human evaluations on sentiment, content preservation, and fluency.

outperforms several approaches that rely on training labels: CrossAligned (Shen et al., 2017) and Delete&Retrieve (Li et al., 2018). However there is still a small gap between our label-free approach and the best-performing labeled model, B-GST (Sudhakar et al., 2019).

In Figure 3, we compare with Lample et al. (2019) on the evaluation setting including pos→pos and neg→neg transfers. Note, this type of evaluation doesn't match our inference procedure, which assumes that the input and output styles differ. Nevertheless, TextSETTR comes fairly close to the performance of Lample et al. (2019), despite not benefiting from training labels.

**Human Evals** As automatic metrics are known to diverge from human judgment (Sudhakar et al., 2019), we conduct human evaluations of the strongest models from Figure 2: TextSETTR, Delete&Retrieve, and B-GST. We sample 200 examples per transfer direction from the Li et al. (2018) test set, and ask three annotators to evaluate each input/output pair on three metrics: sentiment transfer (how well the model changed the sentiment), content preservation, and fluency, on scales of 1–5. Table 1 shows that TextSETTR has superior fluency and performs similarly or better on sentiment, but is worse on content preservation. These results support the view from the automated metrics that TextSETTR achieves similar quality to models that benefit from training labels.

## 3.1 ABLATIONS

**Modifying inference procedure** To better understand the value of our proposed "targeted restyling" mechanism, we consider an alternative inference procedure whereby we ignore the style of the input text, and simply feed the average target exemplar style $v^{trg}$ as the conditioning style vector. Our expectation is that since our learned style space covers multiple attributes, this will have the effect of setting the target attribute (e.g. sentiment), while simultaneously *overwriting* all other style attributes (e.g. formality) with unintended values determined by the average style of the target exemplars. This is borne out in our "overwrite style" ablation, which performs significantly worse than our baseline: accuracy drops from 54.0% to 25.3% while holding self-BLEU constant.

To assess the value of tunable add/delete rates, we also train a model ($-$tunable) without this feature. While the automatic metrics are slightly above the TextSETTR line, we observe several advantages to the tunable model. For one, we observe it significantly reduces the variance in self-BLEU across different inputs. For example, focusing on the case of overly high self-BLEU, we find that without tunable inference, 14.6% of dev eval outputs are identical to their inputs, whereas with tunable inference, this goes to 0.9%. Additionally, through qualitative analysis in Section 4, we find that tunable inference allows more flexibility for controlling different types of transfer.

**Adjusting data sizes** While our unlabeled training data set consists of 23.6M examples, our model only sees 5.1M of these over its 10k steps of training. Yet this is still nearly 10$\times$ more data than the 0.6M examples in the Li et al. (2018) training set used by previous approaches. For a more direct comparison, we experiment with a "small train set", sampling 0.6M examples from our training set. Remarkably, the results in Figure 2 are nearly identical to our baseline, supporting our hypothesis that a fairly lightweight adaptation is sufficient to allow T5 to extract and transfer textual style.

To test the limits of our model's generalization, we reduce the set of exemplars to four manually selected examples of each class.[7] In this setting, we also find reducing delta scale to $\lambda$=4 is beneficial. The results, shown as "manual exemplars" in Figure 2, are still competitive, indicating that our approach generalizes well to the few-shot inference setting. In the other direction, we find that increasing the number of sampled exemplars from 100 to 1000 only provides small additional gains.

---

[7]five stars , amazing . / i really love this product . / works great , will buy again . / it is well made , and a pleasure to use . // zero stars , horrible . / i really hate this product . / does not work, will not buy again . / it is poorly made , and a pain to use .

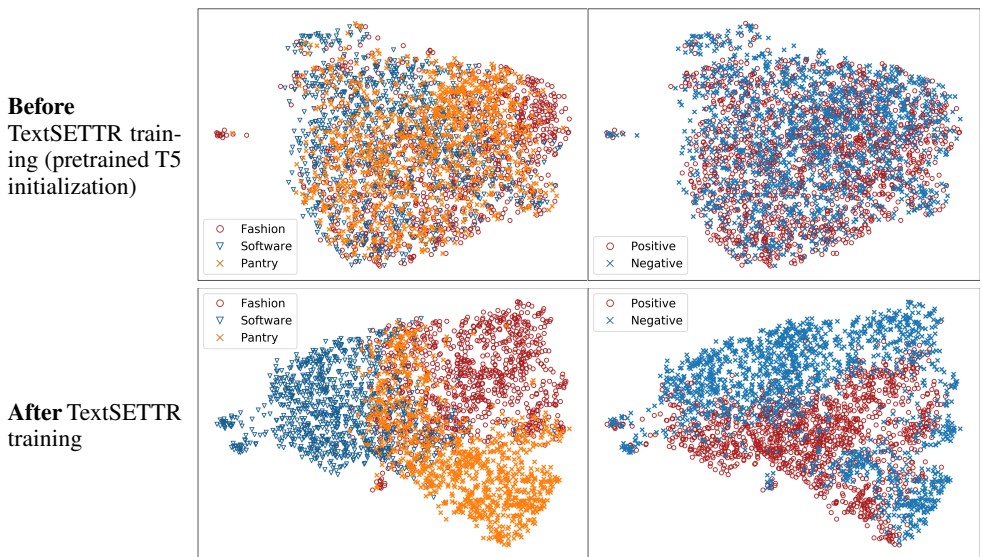

Figure 4: 2D UMAP embeddings of the style vectors extracted by our TextSETTR model before and after training, for text inputs from Amazon reviews covering three product categories and two sentiment labels. Within each row, the same embeddings are visualized with product category labels (left) and sentiment labels (right).

| Reserved ⇒ Emotive | Emotive ⇒ Reserved |
|---|---|
| I liked the movie. | I loved every minute of the movie! |
| ⇒ I cannot even describe how amazing this movie was!! | ⇒ I liked the movie. |
| I was impressed with the results. | I was shocked by the amazing results! |
| ⇒ I was absolutely blown away with the results!! | ⇒ I was surprised by the results. |
| **American ⇒ British** | **British ⇒ American** |
| The elevator in my apartment isn't working. | The lift in my flat isn't working. |
| ⇒ The lift in my flat isn't working. | ⇒ The elevator in my apartment isn't working. |
| The senators will return to Washington next week. | MPs will return to Westminster next week. |
| ⇒ The MPs will return to Westminster next week. | ⇒ Representatives will return to Washington next week. |
| **Polite ⇒ Rude** | **Rude ⇒ Polite** |
| Are you positive you've understood my point? | What the hell is wrong with your attitude? |
| ⇒ you've never understood my point! | ⇒ Perhaps the question is more about your attitude. |
| Could you ask before using my phone? | I could care less, go find somebody else to do this crap. |
| ⇒ I ask you to stop using my phone! | ⇒ I could be wrong, but I would try to find somebody else to do this. |
| **Formal ⇒ Informal** | **Informal ⇒ Formal** |
| I hereby commit to never purchase anything from this institution in the future. | best book ever!! |
| ⇒ i gonna never buy anything from this place again. | ⇒ The book is highly recommended. |
| I couldn't figure out what the author was trying to say. | couldnt figure out what author tryna say |
| ⇒ i dont know what ur trying to say. | ⇒ The reader couldn't figure out what the author was trying to say. |
| **Positive ⇒ Negative** | **Negative ⇒ Positive** |
| I was pretty impressed with the results. | I was pretty disappointed with the results. |
| ⇒ I was pretty disappointed with the results. | ⇒ I was pretty impressed with the results. |
| I will definitely buy this brand again. | I definitely won't buy this brand again. |
| ⇒ I will definitely not buy this brand again. | ⇒ I definitely won't hesitate to buy this brand again. |

Table 2: Examples of transferring along five different axes of style. The same model is used across all examples, with no additional training. Words deleted from the input are red, and words added in the output are blue. Within each category, a fixed tiny set of exemplars is chosen, and fixed delta scale and tuning rates are used. The exemplars and settings are provided in Appendix A.2.

**Modifying training task** Lample et al. (2019) showed promising results by combining noise (N) with back-translation (BT). However we find this combination unstable. When training for 10k steps, our N and N+BT models nearly always copy their input. By increasing the training steps to 50k, we can recover reasonable performance, but the metrics still fall below the TextSETTR line, using our novel NBT task. We also experiment with using NBT in isolation, but this again underperforms our baseline. We expect that the denoising task helps to ensure the NBT inputs (themselves the outputs of denoising) consist of realistic well-formed text. Finally, while Lample et al. (2019) use drop and shuffle noise, we find that only drop and replace are valuable, while shuffle noise can be removed without negative effect.

## 3.2 EMBEDDING VISUALIZATION

To demonstrate that our learned style extractor encodes multiple aspects of textual style, we compute style vectors for 12,000 lines of text from three review categories (Fashion, Software, Pantry) from the Ni et al. (2019) Amazon data. Within each category, we sample 2,000 positives (4 or 5 star) and 2,000 negatives (1 or 2 star), filtering examples where our BERT classifier disagrees with the label. Figure 4 (bottom) plots a 2D UMAP dimensionality reduction (McInnes et al., 2018) of the vectors, and shows clear separations among sentiments and product categories.[8] The top row runs UMAP with the same settings, but over style vectors from our model *before* training, where the style extractor is initialized from pretrained T5. The contrast is a clear indication that our training procedure is helping to learn a representation space where sentiment and topic values are well separated.

## 4 QUALITATIVE ANALYSIS

One advantage of label-free style transfer is that, in theory, a single model can be used to perform transfer along any "dimension" of style, given only a few exemplars, and without the need for additional training. In this section, we investigate the degree to which our approach achieves this goal in practice. For this purpose, we train a single general-purpose TextSETTR model, with the same configuration as our model from Section 3, except fine-tuned for 200k steps on English Common Crawl data (the same "C4" data that T5 pretrained on) as opposed to Amazon reviews.

**Transferable Attributes** Given that our architecture limits the style representation to 1024 dimensions, one may ask how the unsupervised model will make use of this capacity, and which style attributes will be encoded in the learned space. Encouragingly, we find that our model trained on unlabeled Common Crawl data is capable of transferring along many independent axes of style. Table 2 shows selected successful examples of our Common Crawl model transferring emotiveness, dialect, politeness, formality and sentiment. The same model is used in each case, with no additional training. At inference time, a tiny set of exemplars (1–5 examples of each class) is the only labeled data used to compute the style vector delta; these exemplars are presented in Appendix A.2.

Across each type of transfer, we see evidence of generalization beyond the specifics of the chosen exemplars. In making text more emotive, the model uses *amazing* and *blown away*, despite these terms not occurring in the exemplars. In making text more polite, the model inserts novel hedges like *perhaps* and *I could be wrong*. In transferring between American and British styles, the model generalizes to unseen vocabulary items (*elevator ↔ lift*) and draws sound analogies (*senators ↔ MPs*).

**Beyond Style Transfer** We observe three other noteworthy abilities of our model, which we outline briefly here, providing further examples of each in Appendix A.1. First, if inference is tuned to add rather than delete tokens, we find the model is capable of generating style-sensitive **completions**. For instance, if we reuse the American and British exemplars used to produce the results in Table 2 but set the tuning ranges to add:40–70%, delete:0%, the model completes the input *My favorite hot drink:* with either *Starbucks Coffee* (American) or *a mug of tea* (British).

In the other direction, if we tune inference for heavy deletion but apply no modification to the input style vector, we find the model can perform coherent **shortening** of text while leaving most or all

---

[8]We sub-sample to 3,000 points after dimensionality reduction for clearer visualization. Note, we don't expect perfect separation, as inputs may be underspecified for category ("I love this product") or for sentiment ("I bought this last month"). Additionally, since we aim for the learned embedding space to encode many style attributes simultaneously, we don't expect to see crisp linear separation within each attribute.

of the meaning intact. For instance, with add:0–5%, delete:40-90% the model shortens: "They do this without any prior knowledge of cats, for example, that they have fur, tails, whiskers and cat-like faces." ⇒ "They do not know that cats have fur, tails, whiskers and cat-like faces.".

Finally, if instead of applying a *targeted* style delta vector, we add a small *random* delta to the input style, we find our model can output a range of largely meaning-preserving **random augmentations**. For example depending on tuning ranges, the model may transfer the input *What'll the weather be tomorrow?* into outputs *What's the weather forecast for tomorrow?* (add/delete: 30–50%) or *What's the weather like for the next day?* (add/delete: 50-70%). Further examples are in Appendix A.1.

## 5 RELATED WORK

As mentioned at the outset, recent work on text style transfer falls into three classes: supervised, unsupervised, and label-free. Supervised style transfer has seen limited research due to the difficulty of obtaining parallel data. Examples can be seen in Jhamtani et al. (2017) and Carlson et al. (2018).

**Unsupervised Approaches** The bulk of research has focused on "unsupervised" approaches, which rely on labeled but non-parallel data. Typically, labels are assumed to be available for both source and target styles (Dai et al., 2019; He et al., 2020; Lample et al., 2019; Li et al., 2018; Niu et al., 2018; Prabhumoye et al., 2018; Shen et al., 2017; Sudhakar et al., 2019; Wu et al., 2019; Yang et al., 2018). However Zhao et al. (2018) also explore the case where only the target style is labeled. The use of labels at training time can aid modeling, but limits the applicability of these methods, as labeled datasets are not readily available for many attributes of interest.

Our work differs from the above by removing the need for training labels, and offering a single model that can target an unrestricted set of style attributes. Despite these differences, our work shares some similarities with past work. For example, our encoder-decoder architecture and corruption methods are similar to Lample et al. (2019), and we leverage a strong pretrained language model, as in Sudhakar et al. (2019) and Wu et al. (2019).

**Label-Free Approaches** A label-free approach has recently been explored by Xu et al. (2020). The authors train a variational auto-encoder on unlabeled text, where a "manipulable" portion of the latent representation is constrained to fall on a k-dimensional simplex. To perform transfer, they identify empirically the basis vector that most strongly corresponds to the target attribute, and manipulate its magnitude. Compared to our approach, a key difference is that the number of latent factors must be chosen ahead of time, and this limits the number of attributes that may be controlled. Additionally, there is no guarantee that a single basis of the learned simplex will correspond to a target attribute such as dialect or politeness.

**Controlled Generation** A separate strand of research explores "controlled generation" methods for supplementing generative language models to allow control of specific attributes of the output text. As with style transfer, this can be achieved either through labeled examples, as in CTRL (Keskar et al., 2019) and PPLM (Dathathri et al., 2020), or label-free, as in CoCon (Chan et al., 2020). The key difference between these models and style transfer models is that they aim to generate plausible *continuations* following a prompt, as opposed to transferring attributes of a fully-formed input while preserving as much content as possible. It is not clear if controlled generation models could be used to perform style transfer, and they have not to our knowledge been evaluated in this context.

**Tunable Transfer** Our delta scale parameter $\lambda$ is similar to the "modification weight" of Wang et al. (2019) controlling the strength of the transfer operation applied in latent space. However our tunable add/delete rates are novel in offering fine-grain control over the change expected in the *surface* form. This degree of control enables applications such as style-sensitive completion and shortening.

## 6 CONCLUSION

We have presented a unique approach to label-free text style transfer that can achieve results comparable to systems trained with labels (an easier setting), while allowing control of how much of the input is changed. We demonstrate with qualitative results that this approach can produce a single system capable of performing many different types of style transfer, requiring only a handful of exemplars at inference time.

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

# A APPENDIX

## A.1 BEYOND STYLE TRANSFER

In this section, we provide additional examples illustrating the abilities of our TextSETTR model trained on Common Crawl data, beyond typical style transfer.

Examples of **dialect-sensitive completion** are given in Table 3. In each case, the model is passed an input of the form "My favorite X: ". Despite the fact that TextSETTR is not trained specifically for completions, we can use the add and delete rates to encourage the model to insert a few additional tokens, while leaving the original prompt largely unchanged.[9]

The completions show a detailed knowledge of American and British culture. Furthermore, it is remarkable that the model is able to generalize to these "deeper" cultural differences, given only the shallow vocabulary differences presented in the limited set of exemplars in Table 7.

It is also worth highlighting that, thanks to our directional transfer procedure, these completions are not merely "typical American" or "typical British" continuations such as we would expect from a conditional language model trained on each sub-domain of text. Rather, since our inference procedure pushes the style away from one domain and towards the other, the resulting completions are *distinctive* representations of each dialect. As one example, we expect "quinoa" would not only be a common American favorite, but also an uncommon British favorite.

| American ⇒ British | British ⇒ American |
|---|---|
| My favourite food: fish and chips. | My favorite food: quinoa. |
| My favourite hot drink: a mug of tea. | My favorite hot drink: Starbucks Coffee. |
| My favourite dessert: a scone! | My favorite dessert: a brownie. |
| My favourite city: Cardiff. | My favorite city: San Diego. |
| My favourite band: The Beatles. | My favorite band: The Black Keys. |
| My favourite sports league: the English Premier League. | My favorite sports league: the NFL. |
| My favourite newspaper: The Daily Telegraph. | My favorite newspaper: The Washington Post. |
| My favourite museum: the British Museum. | My favorite museum: The National Air and Space Museum. |

Table 3: Examples of dialect-sensitive completion ($\lambda$=8, add:40–70%, delete:0%). In each case, the input text consists of an unfinished phrase, for example: "My favorite food: ". The three exemplars used for each dialect are the same as those used for the transfers in Table 2, as listed in Table 7.

Examples of **shortening** are given in Table 4, with inputs taken from the first five sentences of the Wikipedia article "Artificial neural network". As successful shortening may require minor rephrases, we set our tuning ranges to add:0–5%, delete:40–90%. Since our intention is to leave the textual style unchanged (apart from length), we extract the target style directly from the input text, with no delta added. In each case, the model is largely successful at identifying and removing "superfluous" content, and finding ways of rephrasing to shorten while preserving meaning.

Examples of **random augmentations** are given in Table 5. In each case, we transfer the input sentence "What'll the weather be tomorrow?" to a slightly different style. Specifically, for each transfer, we extract this sentence's style vector and apply a small amount of noise, with each component of the noise vector sampled from a Gaussian $\mathcal{N}(0, 0.08)$. Note that apart from the noise in the style vector, the transfer process is deterministic, as we use greedy decoding.

---

[9]Note however, that in transferring from American to British, the model does prefer to change the prompt from *favorite* to *favourite*.

Artificial neural networks (ANN) or connectionist systems are computing systems that are inspired by, but not identical to, biological neural networks that constitute animal brains.
⇒ Artificial neural networks (ANNs) are computing systems that are inspired by the biological neural networks that constitute animal brains.

Such systems "learn" to perform tasks by considering examples, generally without being programmed with task-specific rules.
⇒ Such systems learn to perform tasks by considering examples, generally without explicit rules.

For example, in image recognition, they might learn to identify images that contain cats by analyzing example images that have been manually labeled as "cat" or "no cat" and using the results to identify cats in other images.
⇒ For example, image recognition systems might learn to identify images that contain cats by analyzing images that have been manually classified as "cat" or "no cat".

They do this without any prior knowledge of cats, for example, that they have fur, tails, whiskers and cat-like faces.
⇒ They do not know that cats have fur, tails, whiskers and cat-like faces.

Instead, they automatically generate identifying characteristics from the examples that they process.
⇒ Instead, they automatically generate identifying characteristics.

Table 4: Examples of shortening (add:0–5%, delete:40-90%), using the first five sentences from the Wikipedia article "Artificial neural network". For each sentence, the target style is extracted directly from the input text, and no delta is added.

The cells of Table 5 apply different tuning ranges, conditioning the model to change a little or a lot. Within each cell, we repeatedly sample the noised style, and present the first five unique outputs. The results indicate that many random changes in style are largely meaning preserving, especially when a small change is requested. With larger add/delete rates, the outputs are still closely related in meaning, despite low lexical overlap.

| Add/Delete: 10–30% | Add/Delete: 30–50% |
|---|---|
| What'll the weather be like? | What's the weather like? |
| What'll the weather be like tomorrow? | What will the weather be like tomorrow? |
| What's the weather like tomorrow? | Will the weather be better tomorrow? |
| What'll the weather be tomorrow? | What's the weather forecast for tomorrow? |
| What's the weather supposed to be tomorrow? | How will the weather be tomorrow? |
| **Add/Delete: 50–70%** | **Add/Delete: 70–90%** |
| Will the weather be perfect tomorrow? | How do you know what the weather will be like? |
| What's the weather for tomorrow? | Is it supposed to be cold tomorrow? |
| What's the weather like on the course? | What will the weather be like in the South? |
| Hopefully the weather will be better tomorrow. | I'm not a fan of the weather. |
| What's the weather like for the next day? | What is the temperature and what is the humidity. |

Table 5: Random augmentations of input text "What'll the weather be tomorrow?", using random style vector deltas with components sampled from $\mathcal{N}(0, 0.08)$.

## A.2 SETTINGS USED FOR QUALITATIVE ANALYSIS

For each of the transfer types (e.g., formal ↔ informal) in Table 2, we specify the intended target styles to the model through a tiny set of exemplars. These exemplars are provided in Tables 6–10. Additionally, for each transfer type, we select a delta scale $\lambda$ and add/delete rates. These settings are selected through initial experiments, and are held fixed across all examples of transfer shown.

## A.3 HUMAN REFERENCE BLEU

Li et al. (2018) provide human reference transfers for their Amazon test data, and report BLEU scores of model outputs against these targets. In principle, we believe this metric is less informative than self-BLEU, as style transfer is a relatively open-ended task, and successful transfers may differ significantly from the single human reference. However, for completeness, we report "reference BLEU" of our models and those of prior work in Figure 5. We observe BLEU and self-BLEU

| Reserved Exemplars | Emotive Exemplars |
|---|---|
| 1. That is a very pretty painting. | 1. OMG, that's such a beautiful painting! |
| 2. I'm excited to see the show. | 2. I'm sooo excited to see the show, it's going to be stellar!! |
| 3. I'm surprised they rescheduled the meeting. | 3. I absolutely can not believe that they rescheduled the meeting! |
| 4. This specimen is an example of the baroque style. | 4. This wonderful specimen is a truly spectacular example of the baroque style. |
| 5. After the performance, we ate a meal. | 5. After the superb performance, we ate a delicious meal. |

Table 6: Emotiveness transfer exemplars. Transfer settings: $\lambda$=9, add/delete rates: 0–100%.

| American Exemplars | British Exemplars |
|---|---|
| 1. It cost ten bucks. | 1. It cost ten quid. |
| 2. My neighbor apologized. | 2. My neighbour apologised. |
| 3. I'm heading out to the bar with some friends. | 3. I'm heading out to the pub with some mates. |

Table 7: Dialect transfer exemplars. Transfer settings: $\lambda$=8, add/delete rates: 10–30%.

| Polite Exemplars | Rude Exemplars |
|---|---|
| 1. No thank you, I'd prefer not to. | 1. Hell no, you can't make me do that. |
| 2. This game could have been better designed. | 2. This game is such a piece of garbage! |
| 3. Do you know why they might have delayed the launch? | 3. Why in god's name would they delay the damn launch? |
| 4. Sorry, I wasn't certain if you were joking. | 4. Are you frigging kidding me? |

Table 8: Politeness transfer exemplars. Transfer settings: $\lambda$=5, add/delete rates: 20–50%.

| Formal Exemplars | Informal Exemplars |
|---|---|
| 1. This was a remarkably thought-provoking read. | 1. reading this rly makes u think |
| 2. It is certainly amongst my favorites. | 2. Its def one of my favs |
| 3. We humbly request your presence at our gala on the 12th. | 3. come swing by our bbq next week if ya can make it |

Table 9: Formality transfer exemplars. Transfer settings: $\lambda$=4, add/delete rates: 40–80%.

| Positive Exemplars | Negative Exemplars |
|---|---|
| 1. Five stars, I love it. | 1. Zero stars, I hate it. |

Table 10: Sentiment transfer exemplars. Transfer settings: $\lambda$=3, add/delete rates: 0–100%.

| Model | BLEU | Self-BLEU |
|---|---|---|
| CrossAligned | 2.0 | 2.9 |
| Delete&Retrieve | 29.7 | 56.9 |
| B-GST | 29.0 | 54.2 |
| CP-G | 17.0 | 35.5 |
| CP-B | 19.4 | 39.8 |
| TextSETTR (0–20%) | 39.0 | 73.3 |
| TextSETTR (10–30%) | 30.7 | 55.8 |
| TextSETTR (20–40%) | 20.0 | 34.7 |
| TextSETTR (30–50%) | 10.6 | 18.4 |
| TextSETTR (40–60%) | 5.5 | 9.1 |
| TextSETTR (50–70%) | 2.2 | 3.6 |

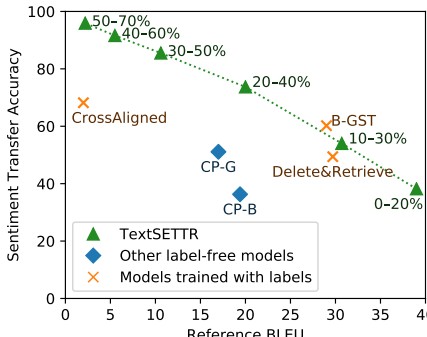

Figure 5: BLEU scores between model outputs and human references provided by Li et al. (2018), along with self-BLEU for comparison. The first group of models in the table had access to labels at training time, while the second group did not. TextSETTR (X–Y%) refers to our model with add/delete rate ranges set to X–Y%.

are highly correlated, and the "Accuracy vs. BLEU" plot conveys the same relationships we saw in Figure 2. As before, all BLEU scores are calculated using SacreBLEU (Post, 2018).

### A.4   AMAZON REVIEWS PREPROCESSING

We use the code in Figure 6 to process raw Amazon reviews from the Ni et al. (2019) dataset and extract pairs of adjacent lines, preprocessed to have a similar format to Li et al. (2018) dataset. We split reviews on newlines, and clip lines to 100 characters, always ending with a period. This gives results similar to Li et al. (2018), where one line may contain multiple sentences, and may consists of a "half-sentence" ending with "e.g." or a similar non-sentence-final period. Additionally, we apply various tokenization and normalization operations to roughly match the observed Li et al. (2018) text.

```python
import re
from html.parser import HTMLParser

html_parser = HTMLParser()

def preprocess(line):
  """Simulate Li et al. preprocessing of one review line."""
  # Lowercase.
  line = line.lower()
  # Replace apostrophes, parens and quotes with spaces.
  line = re.sub("['()\"]", " ", line)
  # Replace dollar values ==> $
  line = re.sub("\$[\d.]*", "$", line)
  # Replace percent values ==> %
  line = re.sub("[\d.]*%", "%", line)
  # Replace single digits ==> num_num
  line = re.sub(" \d[ ,]", " num_num ", line)
  # Replace multi-digits and codes ==> num_extend
  line = re.sub(" \d[^ ]*", " num_extend", line)
  # Remove remaining numbers, including decimals.
  line = re.sub("\d[\d.]*", "", line)
  # Add spaces around certain punctuation marks.
  line = re.sub("([.,?!:])", r" \1 ", line)
  # Remove double spaces after periods before words.
  return re.sub(r"\.  ([a-z])", r". \1", line)

def acceptable_line(line):
  """Check if text looks like an acceptable line from Li et al."""
  if not line or len(line) < 30 or len(line) >= 100:
    return False
  # Avoid lines with any char absent from Li et al. train.
  if re.search('[^ !$%+,.:;>?@\^_`a-z{|}]', line):
    return False
  return True

def clip_to_last_period(line):
  return line[:len(line) - line[::-1].index('.')]

def adjacent_lines(review):
  """Extract a list of adjacent line pairs from review text."""
  review = html_parser.unescape(review)
  review = review.replace('\\"', '"')
  # Simulate Li et al. splitting and filtering.
  if '\n' not in review:
    return
  lines = review.split('\n')
  lines = [preprocess(clip_to_last_period(l[:100]))
           for l in lines if l and "." in l[:100]]
  lines = [preprocess(l) for l in lines]
  lines = [l for l in lines if acceptable_line(l)]
  if len(lines) < 2:
    return
  return list(zip(lines[:-1], lines[1:]))
```

Figure 6: Python code to extract adjacent lines of text from raw Amazon reviews, producing outputs in a similar style to Li et al. (2018).

