# OpenReview forum: "TextSETTR: Label-Free Text Style Extraction and Tunable Targeted Restyling"
_ICLR.cc/2021/Conference — Reject_

### Official Review · AnonReviewer2 · 2020-10-25
**TextSETTR: Label-Free Text Style Extraction and Tunable Targeted Restyling**

**Rating:** 5
**Confidence:** 2

**Review:**

This paper proposes a method for text-style transfer where they dont need label information of the interested style.
The extend t5 model to develop their architecture which models style extract a style vector from arbitrary text and
 use this vector to condition the decoder to perform style transfer.



However the current presentation of the paper is hard to follow. Which raises the following concern:

1. As they need to provide two sentences which has to be chronological sentences, it is not possible to obtain always.
Hence, they randomly select sentence pair, but then two sentences may not bear same style. How the authors are incorporating the same?

2. For inference they need sentence exemplar both both style. This contradicts their previous claim.
They have not compared with

3. How noise introduction in helpful for style corrupted sentence generation? They do not use any heuristic and from a single sentence there can be multiple variation of the corrupted version, are all those taken at the time of training? Then the style it is learning is possibly not the intended one as different corrupted sentences might  need different style to reconstruct the sentence back.

4. Model portion is extremely cryptic. What is back translation etc? At least should be explained in one line.

5. Due to the unreadability of the model, I cannot provide judgement on the result section.

---

> ### Author Response · Authors · 2020-11-17
> **Author response**
>
> We’re sorry that the description of the model wasn’t clear. We’ve provided some clarifications below. Please let us know if there are specific areas in the text that could be clearer.
>
> During training, the sentence preceding the target is used as the style input (seen by the style extractor) and is paired with the corrupted input (seen by the encoder). Perhaps the confusion here is in regards to our BT and NBT tasks, where producing that corrupted input requires first running the model in inference mode. For that inference, we sample a random sentence to use as the style input and in fact hope that it is not of the same style, because we want the corruption to result in a coherent sentence in the wrong style. This results in a training example where the input and output are both coherent sentences in different styles, and the model has to use the style input (which at this point is the actual preceding sentence) to transform the input.
>
> At inference time, it is true that we assume access to exemplars in the source and target styles, despite calling our model label-free. However, we show that even a small number of exemplars is sufficient, and crucially, they don’t need to be chosen before training. Section 4 demonstrates this: we train a single model on a large corpus and can use that model to perform many different types of style transfer. All we need to do to support a new type of transfer is to come up with as few as one exemplar of each style; no additional training of the model is needed.
>
> For noise: While it is true that some of the possible corruptions from random noise may be harder to reconstruct than others, we have no reason to believe that the required style input should be any different. The model is trained to produce an output in the same style as the style input, and the encoder sees a sequence of tokens where most of them are present in the output in roughly the same order, some tokens are missing from the input and need to be introduced by the decoder, and some extraneous tokens are present in the input that need to be ignored by the decoder. Crucially, the style input is not corrupted, and so should always provide a clear signal to guide the decoder.
>
> Back-translation is introduced by Sennrich et al. (2016) “Improving Neural Machine Translation Models with Monolingual Data”. We’ve cited this work, and have a detailed description in Section 2.2. In our case, the method consists of first running the model in inference mode with the original sentence (s_i) as the text input and a random sentence as the style input. This  “corrupted” output text is then fed back as the input text, using the true adjacent sentence (s_{i-1}) as style input, with the goal of reconstructing the original sentence (s_i).
>
> We did another pass through the text and cleared up some of the denser sections.

---

### Official Review · AnonReviewer4 · 2020-10-27
**This paper proposes an approach to label-free style transfer where a noise augmented auto-encoder is conditioned on an input's prior sentence via a "style encoder" and user defined tuning rates with a final objective composed of reconstruction cross entropy and separate noise corruption losses.   The encoders are initialized to a pre-trained T5 weights and at inference time style transfer is performed given examplars of the source and desired target classes for a given input.**

**Rating:** 5
**Confidence:** 4

**Review:**

##########################################################################
Reasons for score:


This paper proposes a novel approach to the label-free style transfer task where an input is corrupted via different strategies and fed into an auto-encoder which is additionally conditioned on its prior adjacent context sentence via a "style encoder" which adds its mean pooled hidden state to the former before decoding.  Both encoders are initialized to and leverage the strength of pre-trained T5 model.  Additionally the amount of addition/deletion of tokens is tunable at both training and inference time.

The overall idea is quite compelling, but the paper's argument could be improved greatly with revisions to its existing experimental setup and more evaluation overall to better and more thoroughly back its claims.


##########################################################################Pros:
Pros:
1) The authors propose a novel approach to the label-free style transfer task that is based on evaluating how training under different combinations of 3 noising strategies  ( Noise, Back Translation and Noisy Back Translation ) on input texts can be used in conjunction with an auto-encoder and style encoder over the prior sentence context to then do inference given an input text and a small number of examplars of the source and target styles.  The idea is laid out fairly clearly both for training and inference though certain particulars there and in the experiments section were a little unclear and could have benefited from some formal notation ( see next section ).

2) The quantitative results on the Amazon dataset for their best model on both the full data and few shot regimes are quite impressive compared with the other label-free style transfer paper they compare against ( Xu 2020 )

3) The few qualitative examples shown are impressive ( particularly the American <-> British ones )

4) The tuning hyper parameter is a useful addition ( though it'd be interesting to see how dataset dependent it is )

##########################################################################
Cons:
1) Overall the writing was a little unclear at certain spots and could have benefitted greatly from some equations explicitly stating the setup.  For instance i was unclear if the context representation was added ( which the text suggests ) or concated to the noisy encoding before being decoded ( the later is suggested by Figure 1 especially since the 4 float values for the tuning rate ranges are said to be prepended ).  Similarly the sampling strategy used ( as opposed to greedy decoding )

2) Doing quantitative evaluation on only one dataset ( Amazon ) and then only showing examples of how the model does qualitatively over another dataset ( English Common Crawl "C4" ) without doing any human eval is a little disappointing.  The idea is novel enough where even just doing some more automated evals would be suffice for me.  For instance, why weren't automated metrics given for the English Common Crawl dataset?  Those results and having information on training set size and the average token size of each example for C4 should be given.  Also the authors compare against Lample 19 for the pos->pos and neg->neg setup for the Amazon data, why not show the results for the SYelp data as well?   Does the 20-40% add/delete tuning work better there as well or is it dataset dependent?

3) There are two issues with your use of the Amazon dataset.  First it doesn't really provide apples to oranges comparison against the prior papers as they train/test on the same data from Li 18 which has ~ 270 K training examples whereas the work here generates 23.6 M training examples.  It seems you should either see how those papers do with that much data or limit your dataset to be of at least comparable size to be fair.   Second, the Amazon test set is only of size 500 so assessing results on that alone seems in-suffice.

4) The paper hypothesizes that style is a "slow moving" feature consistent over large spans of text hence the use of only the prior adjacent sentence as context.  The paper shows that using just an adjacent sentence gives promising results, but doesn't show that its necessarily better than just using examplars or using a leading paragraph to derive the style from.   I don't think this is exactly necessarily to address here, but for future work it would be nice to see such a comparison.  Additionally, how would using a 1000 examplars as opposed to 100 at inference time affect performance?  A graph showing how accuracy and content preservation were affected by that would be interesting to gain better understanding.  Similarly showing how just the NBT strategy did alone (as opposed to N + NBT ) would be interesting.

5) I didn't find the multiple aspect UMAP embedding visualization particularly convincing for how well the embeddings separate the "sentiment" aspect as there is substantial overlap within each category ( particularly software ).  I don't know if this is particularly necessary for your argument in my opinion ( especially compared with evals on other datasets), but if so then it'd be interesting to have quantitative numbers for those separations and compared with how it differs from just taking the T5 embeddings and doing the same UMAP?

6) The "replace" noise strategy feels pretty arbitrary.  Is there any motivation behind using that as opposed to using a LM or another strategy to replace tokens?

7) A citation for using Self Bleu as opposed to Multi-Bleu in the Evaluation Procedure section would be helpful.  Additionally a citation of Ke Wang, Hang Hua, Xiaojun Wan "Controllable Unsupervised Text Attribute Transfer via Editing Entangled Latent Representation" ( Neurips 19 ) particularly for its "tunable" aspect could be an addition to the Related Work section.

8) This is nitpicky and probably for future work, but the use of examplars doesn't necessarily limit the user to a pre-defined set of styles ( like the unsupervised case does ), however it would be interesting to see what would happen given out of domain examplars for either the source or target classes at inference time

##########################################################################

Questions during rebuttal period:

Please address and clarify the cons above

#########################################################################
Possible prior citation

---

> ### Author Response · Authors · 2020-11-17
> **Author response**
>
> Thank you for your thorough review! Here are our initial responses:
>
> 1. Thank you for pointing out these unclear areas. We’ve updated Figure 1 to show that the style vector is added to every encoder output token. We’ve also updated the text around sampling vs. greedy decoding. The BT and NBT sections specify that their corrupted inputs are produced via sampling, while the Evaluation Procedure section specifies that we always use greedy decoding at inference time.
>
> 2. Unfortunately we are not able to experiment on the Yelp data due to legal restrictions. We have **added human evaluations** on sentiment, content preservation, and fluency, covering our model, Delete&Retrieve, and B-GST. To clarify, the C4 (English Common Crawl) dataset is entirely unlabeled, so there isn’t an obvious way to compute automatic metrics on this set. The Li et al. Amazon test sentences are labeled as positive/negative based on the star rating of the review they came from. By contrast, for C4, there are many potential attributes that we could transfer, but no labeled test set for any of them. One of our aims was to demonstrate that our methods extend to *new* style attributes, beyond those for which labeled datasets exist, and using just a few exemplars. This motivated us to show qualitative results on dialect transfer and other types of transfer rather than focus on existing labeled datasets.
>
> 3. You make an excellent point that we train on more data than previous work. For a more fair comparison, we have retrained our model on a random subset of 555K examples, matching the number of examples in the Li et al. 2018 training set used by Xu et al. 2020. Remarkably, the results are nearly identical to our model trained on the 23.6M dataset. This further supports our hypothesis that a fairly lightweight adaptation is sufficient to allow T5 to extract and transfer textual style. We’ve **added this “small train” ablation** to Figure 2 and surrounding discussion. We’ve also clarified that our original models only saw 5.1M examples during the 10k steps of training we use by default. As for the size of the test set, we agree that it is small, but it is the standard evaluation dataset for sentiment transfer and the best point of comparison we can make with prior work.
>
> 4. We have **added an ablation increasing from 100 => 1000 exemplars** as you suggest, and updated Figure 2 and surrounding text. While there is a modest improvement, the relatively small gain indicates that the model already generalizes well from a small set of exemplars. We have also **added an ablation using NBT** alone. This performs relatively poorly compared to the N+NBT baseline, as shown in our updated Figure 2. Our intuition is that the denoising task is helpful to ensure that the NBT inputs (themselves outputs of denoising) consist of realistic well-formed text. We agree that an exciting direction for future work is to explore the use of different/wider contexts for the style input, such as a leading paragraph.
>
> 5. We have **improved the UMAP visualization** by splitting it into two plots, now showing a clearer separation of both sentiment and topic. We also really liked your suggestion of comparing against pretrained T5, so have **added a visualization of pretrained T5 embeddings** in the same figure.
>
> 6. We agree that the replace noise is fairly arbitrary, and could likely be improved on further. But apart from its simplicity, one potential advantage of this replacement strategy is that, unlike a language model, it is likely to insert words that are in the wrong style. This gives the model an opportunity to learn (through the noisy back-translation task) how to detect and modify words that don’t match the target style. That said, we haven’t experimented with LM-based or other more advanced replacement strategies, and it’s an interesting area for more exploration.
>
> 7. Thanks, we’ve **added the citation for Wang et al. (2019)** under Related Work. For the use of self-BLEU, we’re following Sudhakar et al. (2019) and Xu et al. (2020), and have **added citations** under Evaluation Procedure. We focus on self-BLEU primarily to allow a direct comparison with Xu et al. (2020), which is the only other work we’re aware of on *label-free* style transfer. More generally, we believe that self-BLEU is a more informative metric than reference-BLEU (comparing model output to human reference), as many “successful” transfers will differ significantly from the gold reference. However, to allow comparison to work using reference-BLEU, we have **added this metric in the appendix**.
>
> 8. We would like a bit of clarification on this point on what you mean by “out-of-domain”. In the context of sentiment transfer on the Amazon data, would this be positive/negative exemplars that aren’t actually from an Amazon review? Please let us know what you mean; it’s possible that we could perform an experiment before the deadline.

---

### Official Review · AnonReviewer1 · 2020-10-28
**Well motivated method**

**Rating:** 6
**Confidence:** 2

**Review:**

This paper tackles the problem of extracting and modeling text (writing) style without using labeled data.

Traditionally, modeling text style requires either paired sentences (supervised) or two pools of unpaired sentences (so-called unsupervised). This paper exploits 1) language model pretraining and 2) free supervision signals in text to achieve modeling text styles without labeled data.

For the 1st point, the authors (correctly) hypothesize that a large pre-trained language model (e.g. T5) already “knows” about style information and one can isolate the style information using the right fine-tun signal.
For the 2nd point, the authors assume that text style (e.g., sentiment) is slow-moving and consistent for adjacent sentences (I guess It’s a similar signal is exploit by Next Sentence Prediction in BERT, and CBOW in word2vec?). And this is used as the “free” supervision signal to their model finetune.

In the experiments section, the authors test their model on transfer learning tasks. The experiments (Fig 2 &3) seem to suggest that at at given high content preservation score ( > 50), the proposed model is not as accurate as other supervised models. But with low content preservation, the model can steadily improve accuracy by modifier more words (Fig 2).

In Figure 2, the TextSETTR accuracy has almost (inverse) linear response wrt to the content preservation score. But in Figure 3, the plot for TextSETTR stopped at “30-50%”. What would happen if the modification percentage is higher? Would TextSETTR get closer to 100% accuracy?

Another small issue with Figure 3: I believe the task is binary (pos vs neg). It might be more useful to plot the accuracy from 50%~100% instead of from 0% ~100% since 50% is the practical lower bound performance.

---

> ### Author Response · Authors · 2020-11-17
> **Author response**
>
> Thank you for your helpful comments! Here are our initial responses:
>
> Core Assumption: We agree that the use of the preceding sentence as the style input mirrors the Next Sentence Prediction task. We have added a citation for BERT calling attention to this similarity.
>
> Figure 2: Yes, the tuning ranges allow us to trade off between content preservation and accuracy, which is a strength of the model.
>
> Figure 3: Thanks, we’ve added additional data-points to illustrate that our accuracy can reach near 100%, but at the cost of content preservation. We’ve also adjusted the y-axis and added a clarification in the caption that 50% is a lower bound achievable by a trivial model.

---

### Official Review · AnonReviewer3 · 2020-10-28
**The paper proposed a encoder-decoder text style transfer framework without requiring style labels in training.**

**Rating:** 5
**Confidence:** 4

**Review:**

In this paper, the author proposed a transformer-based encoder-decoder framework for label-free text style transfer. The described task under the unsupervised setup is important and instructive for the text style transfer domain. The model architecture is well demonstrated and the writing is easy to follow up. The experiment results show satisfying performance even comparing with state-of-the-art supervised methods.

However, I have some concerns that may lead to the weakness of the paper:

1. About the assumption: The author claimed the method is label-free. However, the "unsupervised" model is based on an assumption that two adjacent sentences should have the same style. With the assumption, the training of the model is actually weak-supervised because in each step the paired sentences are provided with the same style. This assumption is actually utilizing the context-level supervision instead of the sentence-level labels. This idea is also previously used in [1].

2. About the framework: The model adds the exacted style vector to all the hidden states of the encoder. How can the author guarantee that the encoder will not extract the style information of the input? Also, is it possible that the style vectors still contain the content information from context?

3. About the style vector: The model changes the style of the sentence by adding a direction from the source style vector to the target style vector. The approach may work under the assumption that the style vector space is linear to the semantic meanings. But there is no regularizer or training loss to guarantee the linearity assumption of the style extractor. Why didn't the author directly replace the sentence style vector with the target style vector?

4. About the dataset: The model is only evaluated on one dataset. It could be more solid if the author conduct experiment on other commonly-used style transfer datasets such as Yelp and Personality-Captions [2]. Besides, the split Amazon review dataset only has two sentiment classes as "positive" and "negative". It could be more persuasive if the model is tested on other datasets with multiple sentiments, to verify the effectiveness of the proposed re-styling strategy.

5. About the evaluation: The author only reported the performance of content and style preservation (Acc and self-BLEU). The sentence generation quality is expected to report by testing the BLEU score of the generated sentences.

6. In Figure 4, there is no clear pattern between positive and negative sentence embedding. The difference in embedding space is mainly caused by different topics, which in my understanding are the content of sentences. This means the style vectors cannot eliminate the content information and also failed to separate sentences with different sentiments.


Reference:

[1] Zhang et al. Improving the Dialogue Generation Consistency via self-supervised Learning, 2019

[2] Shuster et al. Engaging Image Captioning Via Personality, 2018

---

> ### Author Response · Authors · 2020-11-17
> **Author response**
>
> Thank you for the detailed comments! Here are our responses:
>
> 1. We agree that using sentence adjacency qualifies as a type of weak supervision, and have __added citations__ for both [1] and BERT as prior work. However we feel the term “label-free” is still justified in that no human annotations are needed, unlike previous approaches.
>
> 2. Our model doesn’t place explicit limits on how much style or content each component will extract; instead, we design the training tasks to incentivize them to extract different aspects. Since many of the target tokens will appear in the corrupted input, the model has an incentive to remember the specific input tokens. The style input, however, will have a much lower token overlap with the target and so does not have this incentive; in section 4 we provide evidence that the model is not learning to merely copy words from the exemplars. As for the encoder, if it extracts any style information from the corrupted input, we primarily expect this to just be what is useful for the decoder to identify which input tokens need to change because they are inconsistent with the style input.
>
> 3. “Overwriting” the input style with the average target style would definitely be a simpler inference procedure, so we agree it’s worth an explicit comparison. We have __added this as an ablation__ in the current revision. In practice, we’ve found this performs significantly worse than our proposed procedure. The intuition for why “overwriting” is harmful is discussed in section 2.3. In a nutshell, we only aim to change a single style attribute (e.g. sentiment), while leaving the other style attributes of the input (e.g. formality) unchanged. Using the target style directly would have the effect of setting the target attribute, while simultaneously overwriting all other style attributes to unintended values determined by the average style of the target exemplars.
>
> 4. Besides Amazon, Yelp is the most common style transfer dataset, but unfortunately we are unable to experiment on it due to legal restrictions. The Personality-Captions dataset seems like a great resource for future work. However we could only find two papers [2,3] using this set for evaluating text style transfer (as opposed to controlled generation), and it appears that a standard evaluation protocol hasn’t yet emerged, as these two papers target different styles attributes. Overall, we believe that our Amazon results coupled with our qualitative results on five additional types of style transfer show the versatility of our method.
>
> 5. Thanks, for completeness we have __added BLEU scores__ against the Li et al. gold human references. However, our preference is not to put too much emphasis on these scores, as there are many different ways to adjust the sentiment of an input, and we don’t expect all successful transfers to be close to the human reference. Our main focus is on self-BLEU, which is more directly interpretable and allows direct comparison with Xu et al. (2020), the only other work we’re aware of on *label-free* style transfer.
>
> 6. We have __improved the UMAP visualization__ by splitting it into two plots, now showing a clearer separation of both sentiment and topic. In our view, the separation of topics in the space is a good thing; one of the primary strengths of our model is that it does not require knowledge during training of the intended style(s) to be transferred. The distinction between style (attributes to be changed) and content (attributes to be preserved) is not made until inference time, so the ability of a single model to transfer along many different dimensions of style without further training is only possible because the learned style space encodes many different attributes.
>
> [1] Zhang et al. 2019 Improving the Dialogue Generation Consistency via self-supervised Learning, 2019
>
> [2] Cheng et al. 2020 Improving Disentangled Text Representation Learning with Information-Theoretic Guidance
>
> [3] Li et al. 2020 Complementary Auxiliary Classifiers for Label-Conditional Text Generation

---

### Author Response · Authors · 2020-11-17
**First revision uploaded**

Thank you all for the comments and suggestions!  We’ve uploaded a first revision with the following changes:

1. Split UMAP embedding into two plots (Figure 4, bottom), showing clearer separation of labels. Also added a UMAP embedding using just pretrained T5 (Figure 4, top), showing poor separation between styles.

2. Added four new ablation experiments, as suggested by reviewers. (A) Replace our directional “targeted restyling” with a simpler “overwrite” inference procedure. (B) Increase from 100 => 1000 exemplars. (C) Use noisy back-translation (NBT) as the only training task. (D) Reduce training data to 0.6M, matching the size of Li et al. 2018.

3. Added citations and discussion as suggested. (A) Cite Sudhakar et al. 2019 and Xu et al. 2020 for use of self-BLEU. (B) Cite Zhang et al. 2019 and BERT for leveraging adjacent sentence for weak supervision. (C) Cite Wang et al. 2019 for controlling the “strength” of transfer, while pointing out the differences compared to our tunable add/delete rates.

4. Added more data-points to Figure 3, adjusted y-axis, and mention 50% lower bound.

5. Updates to prose, figures and organization throughout to improve clarity.

We currently have around 1/2 page free, and are planning to make the following additions by the end of this week: (A) Add human evaluation metrics. (B) Add reference-BLEU metrics.

---

### Author Response · Authors · 2020-11-18
**Second revision uploaded**

We’ve uploaded a new revision with the following changes:

1. Updated the Evaluation Procedure section to specify that we use SacreBLEU for calculating self-BLEU scores.

2. In Figure 2, we recalculated content preservation and accuracy metrics for all previous work using our own code. This ensures direct comparability with our models, and corrects for two errors we observed in Sudhakar et al. 2019’s calculations: (i) input/output text is misaligned for Delete&Retrieve and CrossAligned, lowering content preservation scores, (ii) the BLEU calculation is non-standard, splitting on characters instead of words. We compute accuracy using our BERT classifier across the board, again making the comparisons more meaningful. The changes are visible in Figure 2. The overall ranking of models is the same, but the gain of our model over previous label-free work (CP-G, CP-B) is clearer, and the gap between our work and the best model using labels (B-GST) is considerably smaller.

We are still planning to make the following additions by end of week: (A) Add human evaluation metrics. (B) Add reference-BLEU metrics.

---

### Author Response · Authors · 2020-11-20
**Third revision uploaded**

We’ve uploaded a third revision with the following additional changes:

1. Added human evaluation metrics and discussion at the end of the Experiments section. The results support the view from the automated metrics that TextSETTR achieves similar quality to the strongest models that benefit from training labels.

2. Added reference-BLEU metrics and discussion in Appendix A.3, as well as a brief discussion in the Evalation Procedure section of the relation between reference-BLEU and self-BLEU.

---

### Author Response · Authors · 2020-11-24
**Overall author response**

We thank all the reviewers for their time and thoughtful feedback.

The reviewers with higher confidence (R3, R4) pointed out several core strengths of the paper:

R3 writes that the unsupervised setup we describe is **“important and instructive”**, the model architecture is **“well demonstrated”**, and the writing is **“easy to follow”**. They also observe that our label-free results show satisfying performance **“even comparing with state-of-the-art supervised methods”**.  (Note, our current results are now *even stronger* after addressing an error in Sudhakar et al. 2019’s calculations; see below.)

R4 writes that we present a **“novel approach”**, and that the overall idea is **“quite compelling”** and **“laid out fairly clearly”**. They observe that our quantitative results are **“quite impressive”** compared with previous label-free work, and also find our qualitative examples **“impressive”**, particularly the American <-> British ones.

At the same time, R3 and R4 hoped to see a more thorough evaluation, and neither were convinced by our original UMAP visualization of the learned representation space. Additionally, R2 and R4 found some points unclear and asked for more clarity in the presentation, while R1 suggested changes to two of our figures.

The current revision includes significant improvements to evaluation, analysis and clarity, all grounded in reviewer feedback. The highlights are:

1. **Strengthened evaluation** (A) Added human evaluation results covering our model and several competitors, which support the view from automated metrics. (B) Added reference-BLEU metrics, which we find correlate strongly with self-BLEU. (C) Recalculated BLEU metrics and sentiment accuracy scores for previous work directly from their text outputs, to ensure consistency and address errors we observed in Sudhakar et al. 2019’s calculations; see our “Second revision uploaded” comment for full details.

2. **Added four new ablations** suggested by reviewers. (A) Replace our directional “targeted restyling” with a simpler “overwrite” inference procedure. (B) Increase from 100 => 1000 exemplars. (C) Use noisy back-translation (NBT) as the only training task. (D) Reduce training data to 0.6M, matching the size of Li et al. 2018.

3. **Improved UMAP analysis** (A) Split the UMAP visualization of our style vectors into two plots (Figure 4, bottom), showing much clearer separation of labels. (B) Added a visualization of *pretrained* T5 embeddings for comparison (Figure 4, top), showing the value of our proposed fine-tuning procedure.

See our responses below for more detailed description of the changes. We have also responded to individual reviews directly in the “Author Response” comments below.

---

### Decision · Program_Chairs · 2021-01-07
**Final Decision**

**Decision:**

Reject

**Comment:**

This paper proposes a new method for label-free text style transfer. The method employs the pre-trained language model T5 and makes an assumption that two adjacent sentences in a document have the same style. Experimental results show satisfying results compared with supervised methods.

Pros. • The paper is generally clearly written. • The proposed method appears to be new. • Experiments have been conducted.

Cons • The fundamental assumption of the method is not convincing enough. (Issue 1 of R3, Issue 4 of R4, Issue 1 of R2) • The proposed model is also not convincing enough. (Issues 2 and 3 of R3, Issue 3 of R2) • There are problems with the experiments. For example, it would be better to use more datasets in the experiments. (Issue 4 of R3, Issue 2 of R4)

Discussions have been made among the reviewers. The reviewers appreciate the efforts made by the authors in the rebuttal, including the additional experiments. However, they are not fully convinced and still feel that the submission is not strong enough as an ICLR paper.